# Genome-Wide Association Analysis and Genetic Parameters for Feed Efficiency and Related Traits in Yorkshire and Duroc Pigs

**DOI:** 10.3390/ani12151902

**Published:** 2022-07-26

**Authors:** Weining Li, Zhaojun Wang, Shenghao Luo, Jianliang Wu, Lei Zhou, Jianfeng Liu

**Affiliations:** 1College of Animal Science and Technology, China Agricultural University, Beijing 100193, China; liwn@cau.edu.cn (W.L.); luoshenghao97@163.com (S.L.); 2Beijing Zhongyu Pig Breeding Co., Ltd., Beijing 100194, China; wangzhaojun@vip.sohu.com (Z.W.); wwwjl1617@163.com (J.W.)

**Keywords:** feed efficiency, pig, GWAS, genetic parameters, DEBVs

## Abstract

**Simple Summary:**

Genetic improvements in feed efficiency (FE) and related traits could considerably reduce pig production costs and energy consumption. Thus, we performed a genetic parameter estimation and genome-wide association study of four FE and FE-related traits, namely, average daily feed intake, average daily gain, the feed conversion ratio, and residual feed intake, of two pig breeds, Yorkshire and Duroc. The results demonstrate the genetic relationships of FE and FE-related traits with two growth traits, age and backfat thickness at 100 kg. We also identified many single-nucleotide polymorphisms (SNPs) and novel candidate genes related to these traits. In addition, we found many pathways significantly associated with FE and FE-related traits, and they are generally involved in digestive and metabolic processes. The results of this study are expected to provide a valuable reference for the genomic selection of FE and FE-related traits in pigs.

**Abstract:**

Feed efficiency (FE) traits are key factors that can influence the economic benefits of pig production. However, little is known about the genetic architecture of FE and FE-related traits. This study aimed to identify SNPs and candidate genes associated with FE and FE-related traits, namely, average daily feed intake (ADFI), average daily gain (ADG), the feed conversion ratio (FCR), and residual feed intake (RFI). The phenotypes of 5823 boars with genotyped data (50 K BeadChip) from 1365 boars from a nucleus farm were used to perform a genome-wide association study (GWAS) of two breeds, Duroc and Yorkshire. Moreover, we performed a genetic parameter estimation for four FE and FE-related traits. The heritabilities of the FE and FE-related traits ranged from 0.13 to 0.36, and there were significant genetic correlations (−0.69 to 0.52) of the FE and FE-related traits with two growth traits (age at 100 kg and backfat thickness at 100 kg). A total of 61 significant SNPs located on eight different chromosomes associated with the four FE and FE-related traits were identified. We further identified four regions associated with FE and FE-related traits that have not been previously reported, and they may be potential novel QTLs for FE. Considering their biological functions, we finally identified 35 candidate genes relevant for FE and FE-related traits, such as the widely reported *MC4R* and *INSR* genes. A gene enrichment analysis showed that FE and FE-related traits were highly enriched in the biosynthesis, digestion, and metabolism of biomolecules. This study deepens our understanding of the genetic mechanisms of FE in pigs and provides valuable information for using marker-assisted selection in pigs to improve FE.

## 1. Introduction

Feed costs account for approximately 64–72% of the total cost of pig production [1]. Improving the feed efficiency (FE) of pigs dramatically reduces production costs. The feed conversion ratio (FCR) and residual feed intake (RFI) are two traits that have been used to evaluate feed efficiency [2]. Studies have shown significant genetic correlations between feed efficiency traits with average daily gain (ADG) and average daily feed intake (ADFI) [3,4], so they are important indicators of feed efficiency. In addition to nutritional and management strategies [5,6], the identification of causative mutations constitutes a promising perspective for the application of genetic selection on FE relying on molecular-based prediction. An effective genome-based breeding program requires the understanding of the genetic architecture of these traits. Genome-wide association analysis (GWAS) has previously been demonstrated to be an effective method for the detection of FE and related quantitative trait loci (QTL) and candidate genes in pigs. A previous study showed that a total of 829 selected single-nucleotide polymorphisms (SNPs) from an association analysis explained 61% of the phenotypic variance in RFI [7]. Wu et al. [8] identified 24 SNPs that had direct genetic effects and 31 SNPs that had social genetic effects on ADFI and ADG in Yorkshire pigs. Another study identified eight common significant QTL regions associated with the same four FE and FE-related traits as those in our study [9]. Do et al. performed a GWAS, and 19 SNPs were found to be significantly associated with two different measures of RFI based on the phenotypes of 596 Yorkshire boars. In addition to SNPs, studies have also revealed many potential candidate genes associated with FE and FE-related traits in pigs. The melanocortin 4 receptor (*MC4R*), a signaling molecule involved in the regulation of energy homeostasis [10], was identified as a candidate gene associated with ADG, ADFI [11,12], and FCR [13]. Duy et al. [14] found that several nearby genes of significant SNPs for RFI were clustered in the insulin signaling pathway, and they finally identified insulin receptor tyrosine kinase (*INSR*) as a candidate gene for RFI. Several genes involved in the transport process, such as *AQP4*, *SLC22A23*, and *SLC6A14*, have been identified as potential candidate genes for FCR and ADFI in previous studies [15]. An enrichment analysis showed that candidate genes affecting FCR pointed to corresponding biological pathways related to lipid metabolism, olfactory reception, and immunological status [16]. These studies might contribute to the improvement of the genome selection of FE and FE-related traits. However, only 454 and 385 QTLs have been listed as feed conversion and feed intake traits, respectively, while more than 2597 and 3511 QTLs have been listed as growth traits and fatness traits, respectively (PigQTLdb, https://www.animalgenome.org/cgi-bin/QTLdb/SS/index (accessed on 13 May 2022)). Studying genetic structures and variants affecting FE and FE-related traits does not provide enough detail since phenotype data collection is difficult. Fu et al. [9] found that three QTL regions related to ADFI and RFI traits overlapped. Ding et al. [17] found that a key SNP on SSC 7 contributed 2.16% and 2.37% of the observed phenotypic variance for DFI and RFI, respectively, and that another key SNP on SSC 1 contributed 3.22% and 5.46% of the observed phenotypic variance for FCR and RFI, respectively. Additionally, many studies have found significant correlations of FE and FE-related traits with growth traits [3,4,18]. Identifying genetic correlations with main growth traits is required before FE and FE-related traits can be included in the breeding program.

To further identify quantitative trait loci (QTL) and potential genes associated with four FE and FE-related traits in pigs, we conducted a GWAS on two purebred pig breeds, Yorkshire (YY) and Duroc (DD). In addition, we estimated the genetic parameters of FE and FE-related traits and their genetic correlation with two growth traits, backfat thickness at 100 kg (BF) and age at 100 kg (AGE).

## 2. Materials and Methods

### 2.1. Phenotypes and Genotypes

The data used in this study were all from a nucleus breeding farm in the province of Inner Mongolia of China. Details of the phenotype data collection and processing procedures are explained in Appendix B. After data cleaning, a total of 3661 YY boars were recorded to have at least one of the four FE and FE-related traits or two growth traits, and 239,234 individuals were included in the pedigree, including 880 boars genotyped with an in-house-designed SNP chip named CAU50K (detailed information presented in Appendix C). For DD, there were 2162 boars recorded with at least one of the six traits in this study, and 24,576 individuals were included in the pedigree, of which 485 boars were genotyped with the CAU50K SNP chip. After quality control, YY and DD populations contained 32,322 and 25,539 SNPs, respectively.

### 2.2. Statistical Model

The best linear unbiased prediction (BLUP) method was applied to the six traits in this study to determine the (co)variance components. For an evaluation of the genetic relationships, the phenotypes have eight combinations between two growth traits with four FE and FE-related traits. The following univariate and bivariate animal models were used to estimate the variance components:$$y=Xb+Z_{1}a+Z_{2}p+e$$
where ***y*** is the vector of observations for one of the six performance traits in univariate analyses, while in the bivariate model, ***y*** is the vector of observations composed of one FE-related trait and one growth trait. b is the vector of the fixed year-season effect; a is the vector of additive genetic effects; p is the vector of random litter effects; e is the vector of residual effects; and X, $Z_{1}$, and $Z_{2}$ are the incidence matrices of b, a, and p, respectively. The distributions assumed for the random terms in the univariate model are $a∼N\left(0,\;A\sigma_{a}^{2}\right)$, $p∼N\left(0,I\sigma_{p}^{2}\right)$, and $e∼N\left(0,I\sigma_{e}^{2}\right)$, while in the bivariate model, they are $a∼N\left(0,\;G_{0}⊗A\right)$, $p∼N\left(0,R_{p}⊗I\right)$, and $e∼N\left(0,R_{e}⊗I\right)$. $\sigma_{a}^{2}$, $\sigma_{p}^{2}$, and $\sigma_{e}^{2}$ are the additive genetic, litter, and residual variances, respectively. $G_{0}$, $R_{p}$, and $R_{e}$ are the 2 × 2 symmetrical direct additive genetic effect, litter effect, and residual effect (co)variance matrices, respectively. ***I*** denotes the identity matrix of adequate dimension, ***A*** denotes the numerator relationship matrix, and ⊗ denotes the Kronecker product. The average body weight (ABW) of each individual was computed as the average of the body weight at the start and end of testing. Metabolic body weight (MBW) mid-test calculated as ABW raised to the power of 0.75 [17] was included as a covariate for ADFI and FCR.

Using the estimated variance or covariance component from the bivariate analysis, estimated genetic correlations were calculated as $r_{g}=\sigma_{a_{12}}/\left(\sigma_{a_{1}}\sigma_{a_{2}}\right)$. We performed a chi-square test with one degree of freedom using genetic correlation estimates and their SE to study whether the genetic correlation is different from 0.

In the GWAS, a single-step genomic BLUP (ssGBLUP) [19,20] was first carried out to estimate the genomic breeding values (GEBVs) for all animals in the pedigree by combining their pedigree and genomic information, and the inverse of the relationship matrix (***H***) was as follows:$$H^{-1}=A^{-1}+\left[\begin{array}{cc} 0 & 0\\ 0 & G^{-1}-A_{22}^{-1} \end{array}\right]$$
where $G^{-1}$ is the inverse of the genomic relationship matrix, and $A_{22}^{-1}$ is the inverse of the numerator relationship matrix for genotyped individuals. ***G*** was calculated as $G=wGr+\left(1-w\right)\;A_{22}$ [21], where w = 0.95, and ***Gr*** is a genomic matrix before weighting, calculated as VanRaden [22]. The same model as the above univariate model was used for GEBV estimation by replacing ***A*** with ***H***. Estimated variance components/GEBVs were determined based on BLUP and ssGBLUP using the DMUAI module in the DMU software package e. To eliminate the contribution of information from relatives, de-regressed estimated breeding values (DEBVs) were used as the response variables in the GWAS. According to the method proposed by Garrick et al. [23], the DEBVs of the genotyped individuals were calculated based on the GEBVs from the univariate model using the R package “blupADC” [24]. The GWAS was performed with the following a single-marker regression model using GEMMA Software [25]:y=Xm+Wa+e
where ***y*** is the vector of the dependent variable (DEBV); ***m*** is the vector of the SNP marker effects; a is the vector of the residual polygenic effects with a normal distribution $a∼N\left(0,G\sigma_{a}^{2}\right)$, where ***G*** is the realized relationship matrix constructed with markers [8], and $\sigma_{a}^{2}$ is the additive genetic variance; ***e*** is the vector of residual errors with a normal distribution $e∼N\left(0,\;I\sigma_{e}^{2}\right)$, where ***I*** is an identity matrix, and $\sigma_{e}^{2}$ is the residual variance; and ***X*** and ***W*** are the incidence matrices of ***y*** and are related to ***m*** and ***a***, respectively.

For each SNP, the Wald statistic [26,27] was implemented to examine the significance of the SNP’s association with each trait. Then, the Bonferroni correction was implemented to define the significant threshold. To avoid missing the true hints of linkage, the genome-wide significant and suggestive levels were set as *p* = 0.05/N and *p* = 1/N, respectively, where N is the number of analyzed SNPs [28].

### 2.3. Candidate Genes and Functional Analysis

PigQTLdb [29] was used to annotate significant SNPs located in previously mapped QTLs in pigs. The definition of QTL intervals followed the same procedures as those of Delpuech et al. [30]. The region within 1 Mbp centered on each significant SNP was defined as a “QTL window”. Then, overlapping windows were combined into a single QTL region per trait within each breed. Gene contents within the detected QTLs were retrieved from the Ensembl Genes 105 Database based on the Sscrofa11.1 genome assembly using BioMart [31]. The biological functions of these genes were investigated in GeneCards [32] and the reported literature. To provide insight into the functional enrichment of genes, Gene Ontology (GO) [33] term and Kyoto Encyclopedia of Genes and Genomes (KEGG) [34] pathway enrichment analyses were subsequently conducted using the KOBAS [35] database.

## 3. Results and Discussion

### 3.1. Phenotypes and Estimated Heritabilities

Table 1 shows the descriptive statistical results of the phenotypes and heritabilities estimated in this study. A two-tailed Student’s *t*-test showed that the phenotypes among the breeds were significantly different for all six traits. In general, AGE has a moderate heritability, whereas BF has a moderate-to-high heritability [4,36]. In this study, AGE and BF also had moderate heritabilities in YY and DD pigs (Table 1). The heritabilities of the four traits in this study ranged from 0.14 (RFI) to 0.24 (ADFI) for YY and from 0.13 (ADG) to 0.36 (FCR) for DD. The heritabilities estimated in this study were lower than those in the study conducted by Homma et al. [18], which did not consider random environmental effects. Our results agree with those of a previous study [4]; the estimated heritabilities are different among breeds. In Chinese pig breeding, DD pigs are generally used as terminal sires in combination with crossbred Landrace x Yorkshire sows. In DD pigs, there is a greater focus on the selection of growth and FE, while in YY pigs, a major emphasis is placed on improving litter size. In our study, the heritabilities of FE and FE-related traits in DD were all above 0.30, except for that of ADG. Therefore, differences in breed management might lead to differences in estimated heritability. In addition, differences in the inherent genetic background of the varieties may be the main reason for this. Overall, it can be concluded that the heritability estimates for the FE and FE-related traits of pig breeds vary considerably depending on trait, statistical model, and population. The results show that these traits have a moderate heritability. Thus, FE and FE-related traits could gain considerable genetic advances with the implementation of genomic selection.

### 3.2. Genetic Correlations of FE and FE-Related Traits with Growth Traits

Except for BF and RFI in DD (0.07), non-ignorable genetic correlations (>0.10) can be found between two growth traits with FE and FE-related traits (Table 2). However, it is worth noting that several genetic correlation coefficients have relatively large standard errors. Consistent with Do et al. [4], we found that the BF of YY had positive genetic correlations with ADFI (0.44) and ADG (0.13), while it was negative with RFI (−0.14). This result implies that the selection of BF might have a negative effect on these FE and FE-related traits, which is an unfavorable phenomenon for breeders. However, Hong et al. [37] and Homma et al. [18] obtained positive estimated genetic correlations between BF and RFI. Surprisingly, AGE was negatively correlated with ADFI (−0.25 for YY and −0.36 for DD) and ADG (−0.69 for YY and −0.56 for DD), implying that selecting for a lower AGE can improve a pig’s feed intake and growth rate. We also found that AGE and BF had positive genetic correlations with FCR, suggesting that selecting a lower AGE made animals eat more slowly, and this has also previously been reported [37]. As discussed above, breeding experts must carefully determine the breeding scheme according to the genetic relationships of growth traits with FE and FE-related traits in order to obtain the maximum benefit.

### 3.3. Genome-Wide Association Study

The descriptive statistics of the DEBVs used for the GWAS for FE and FE-related traits included in this study are shown in Appendix A. We identified 61 significant SNPs (Appendix A) located on 12 different chromosomes (SSC), of which 15 SNPs reached genome-wide significant thresholds (Table 3). As in previous studies [4,17], the genetic backgrounds of the FE and FE-related traits were significantly different between breeds. In addition, the *P*-values of the GWAS results were visualized using Manhattan plots and Q–Q plots (Figure 1 and Figure 2) by the R package “CMplot” [38]. Most of the significant SNPs were retrieved in PigQTLdb (Appendix A), revealing the reliability of the results. Surprisingly, we found one novel SNP significantly related to each trait, ADFI (SSC18: 32623286), ADG (SSC11: 67792739), FCR (SSC15: 16281234), and RFI (SSC8: 89446476), which might be a potential new causative mutation for FE and FE-related traits.

### 3.4. Candidate Genes and Functional Analysis

After merging overlapping “QTL windows”, we finally identified 8, 20, 2, and 7 QTLs associated with ADFI, ADG, FCR, and RFI, respectively (Appendix A). The four QTLs determined by four novel significant SNPs have not been reported in PigQTLdb. Considering the biological functions of the genes annotated based on QTLs, 35 genes were identified as positional candidate genes for FE and FE-related traits (Appendix A). For example, ATPase phospholipid transportation 8B1 (*ATP8B1*), a candidate gene of ADFI, is a protein coding gene that is involved in ion-channel transport and the transport of glucose, bile salts, organic acids, metal ions, and amine compounds, and it has been identified as a candidate gene for backfat thickness in pigs [39]. The melanocortin 4 receptor (*MC4R*) and lectin, mannose binding 1 (*LMAN1*) were also identified as candidate genes for ADFI, and they have been reported to be associated with AGE [10,11,12,13] and ADFI [40], respectively. The insulin receptor (*INSR*) has repeatedly been reported to be an important gene affecting RFI in relevant studies [14,41,42] because it plays an important role in synthesizing and storing carbohydrates, lipids, and proteins. In addition, we identified eight candidate genes (Appendix A) in four novel QTLs, increasing the credibility of these QTLs, such as *ZRANB3* and *MCM6*, which are essential for the initiation of genome replication.

In total, 617 enriched GO terms and 102 KEGG pathways were identified for candidate genes of FE and FE-related traits, of which, 96 GO terms and 4 KEGG pathways were considered statistically significant at FDR-corrected *p* < 0.05. The top 30 enrichment terms are shown in Appendix A. The GWAS signals for FE and FE-related traits were highly enriched in the biosynthesis (GO:0017101), digestion (KEGG:ssc04973), and metabolism (KEGG:ssc00052) of biomolecules, such as carbohydrates and proteins. These results are consistent with earlier findings by Li et al. [43]. We found that sialylation was significantly associated with ADFI (GO: 0097503; *p* < 0.01), which is involved in the covalent connection between sialic acid and substrate molecules and plays an important role in the metabolism of macromolecular organic substances. The ephrin receptor signaling pathway (GO: 0019899; *p* < 0.01) enriched on ADG is very important for protein synthesis, and it regulates many major cellular processes that produce or use large amounts of energy and nutrients. Therefore, it is an essential pathway for FE and FE-related traits [44]. Overall, FE and FE-related traits were found to be related to diverse biological processes and pathways, suggesting that they are highly polygenic traits regulated by many genes.

## 4. Conclusions

In conclusion, the results show that the heritabilities of FE and FE-related traits ranged from 0.13 to 0.36, and there were distinct genetic correlations (−0.69–0.52) of growth traits with FE and FE-related traits. We identified four critical genomic regions for ADFI (SSC18: 32.12–33.12 Mbp), ADG (SSC11: 67.29–68.29 Mbp), FCR (SSC15: 15.78–16.78 Mbp), and FCR (SSC8: 88.95–89.95 Mbp), and they might be potential new QTL regions for FE. We also further identified 35 potential candidate genes for four FE and FE-related traits, and they play an essential role in many biological processes, such as ion transfer, mitochondrial activities, and the macromolecule metabolism. Some interesting KEGG pathways and GO terms, e.g., sialylation, were found to have potential functions in FE in pigs. Overall, this study’s estimation of genetic parameters and GWAS results helps us better understand the genetic structure of FE and FE-related traits and provides important information for the future genomic predictions of these traits.

## Figures and Tables

**Figure 1 animals-12-01902-f001:**
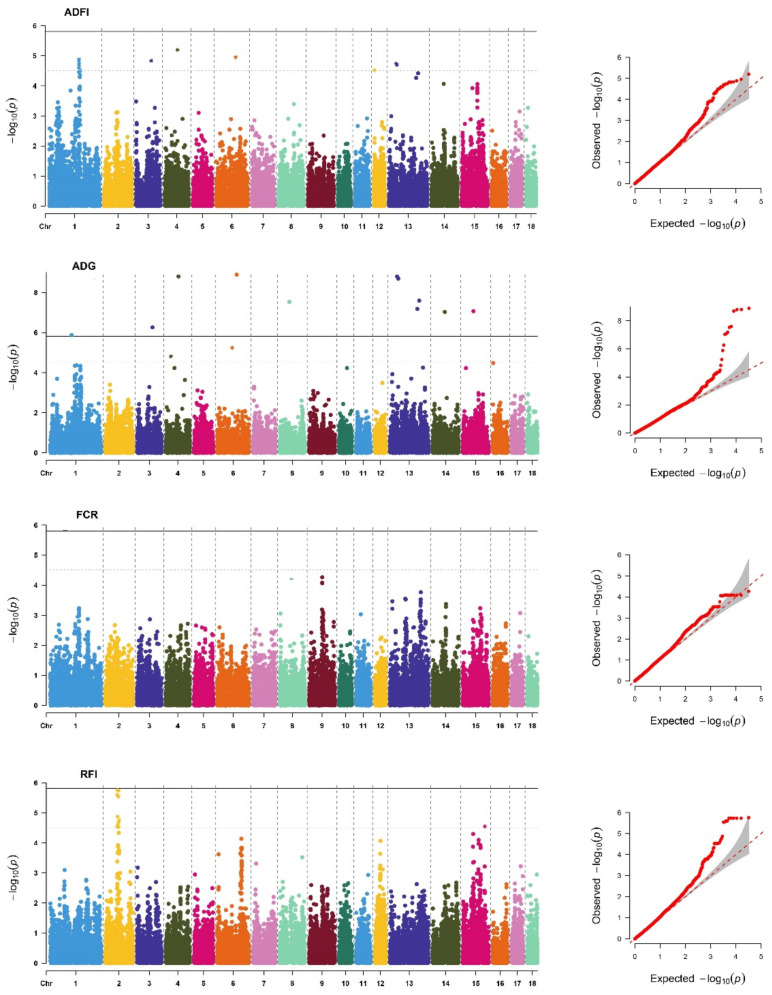
Manhattan plots and Q-Q plots of SNP additive effects for average daily feed intake (ADFI), average daily gain (ADG), feed conversion ratio (FCR), and residual feed intake (RFI) traits of Yorkshire. The *X*-axis is the position of SNP on each chromosome, and the *Y*-axis is the significant level (−log_10_
*p*-value). The solid line indicates genome-wide significance (*p*-value = 1.55 × 10^−6^), and the dashed line shows suggestive significance with a *p*-value threshold of 3.09 × 10^−6^.

**Figure 2 animals-12-01902-f002:**
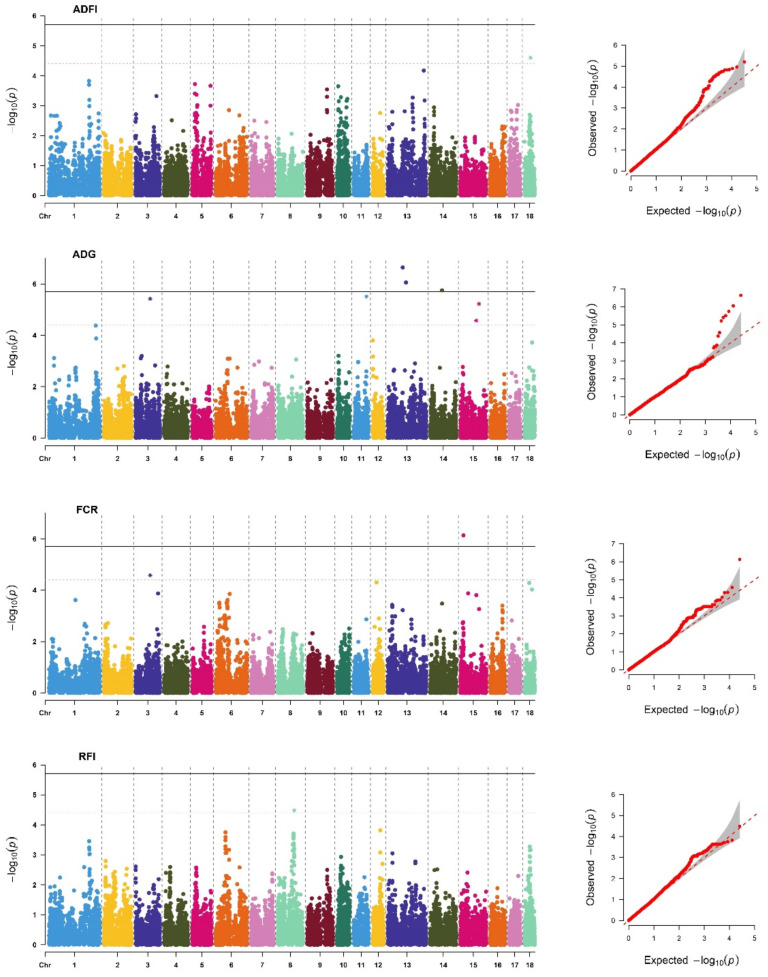
Manhattan plots and Q-Q plots of SNP additive effects for average daily feed intake (ADFI), average daily gain (ADG), feed conversion ratio (FCR), and residual feed intake (RFI) traits of Duroc. The *X*-axis is the position of SNP on each chromosome, and the *Y*-axis is the significant level (−log_10_
*p*-value). The solid line indicates genome-wide significance (*p*-value = 1.96 × 10^−6^), and the dashed line shows suggestive significance with a *p*-value threshold of 3.92 × 10^−6^.

**Table 1 animals-12-01902-t001:** Number of observations (N), means (standard deviations), and heritabilities (standard errors).

Traits	Unit	YY			DD		
		N	Means ± SD ^1^	*h*^2^ ± SE	N	Means ± SD	*h*^2^ ± SE
ADFI	kg/d	3656	2.38 ± 0.35 ^a^	0.24 ± 0.04	2156	2.76 ± 0.35 ^b^	0.36 ± 0.08
ADG	kg/d	3555	0.83 ± 0.08 ^a^	0.16 ± 0.04	2156	0.84 ± 0.08 ^b^	0.13 ± 0.06
FCR	kg/kg	3629	2.56 ± 0.24 ^a^	0.17 ± 0.04	2148	2.70 ± 0.25 ^b^	0.31 ± 0.07
RFI	kg	3556	−0.07 ± 0.17 ^a^	0.14 ± 0.03	2154	0.03 ± 0.19 ^b^	0.32 ± 0.07
BF	mm	3557	12.23 ± 2.46 ^a^	0.48 ± 0.05	2157	11.75 ± 2.06 ^b^	0.36 ± 0.07
AGE	d	3548	159.88 ± 13.1 ^a^	0.38 ± 0.06	2148	149.32 ± 9.7 ^b^	0.19 ± 0.06

^1^ Means with different letters in a row are significantly different according to two-tailed Student’s *t*-test (*p* < 0.01).

**Table 2 animals-12-01902-t002:** Genetic correlations (standard errors) among FE-related traits and growth traits in Yorkshire (upper triangle) and Duroc (lower triangle) pigs.

Traits	ADFI	ADG	FCR	RFI	BF	AGE
ADFI		0.62(0.11)**	0.41(0.14) **	0.80(0.06) **	0.44(0.10) **	−0.25(0.19)
ADG	0.45(0.17) **		−0.44(0.13) **	0.04(0.18)	0.13(0.13)	−0.69(0.10) **
FCR	0.56(0.14) **	−0.44(0.19) *		0.48(0.12) **	0.52(0.11) **	0.38(0.18) *
RFI	0.93(0.04) **	0.18(0.24)	0.77(0.08) **		−0.14(0.19)	0.32(0.13) *
BF	0.34(0.14) *	−0.13(0.24)	0.23(0.16)	0.07(0.17)		−0.18(0.11)
AGE	−0.36(0.21)	−0.56(0.28) *	0.27(0.23)	−0.11(0.22)	0.15(0.21)	

* (*p* < 0.05) and ** (*p* < 0.01) indicate genetic correlations different from zero with the chi-square test.

**Table 3 animals-12-01902-t003:** Identification of genome-wide significant SNPs associated with FE-related traits in pigs.

Breeds	Traits	Chr	Location(bp)	SNP Name	Alleles ^1^	MAF ^2^	*p*-Value
DD	ADG	13	80,501,143	seq-rs710999761	T/G	0.468	2.27 × 10^−7^
		13	98,302,557	seq-rs334871208	T/C	0.295	8.75 × 10^−7^
		14	64,144,092	seq-rs80921027	A/G	0.282	1.77 × 10^−6^
	FCR	15	16,281,234	seq-rs329844461	C/T	0.419	7.36 × 10^−7^
YY	ADG	1	115,356,348	seq-rs344383954	C/A	0.050	1.32 × 10^−6^
		3	83,351,277	seq-rs334252973	C/T	0.294	5.46 × 10^−7^
		4	69,687,124	seq-rs322234522	T/C	0.053	1.58 × 10^−9^
		6	105,104,215	seq-rs320347867	A/G	0.053	1.29 × 10^−9^
		8	53,757,344	seq-rs339132738	T/C	0.056	2.92 × 10^−8^
		13	38,267,479	seq-rs705817794	A/G	0.052	1.60 × 10^−9^
		13	44,606,060	seq-rs793013452	C/A	0.050	2.03 × 10^−9^
		13	147,609,391	seq-rs705621029	A/C	0.056	6.61 × 10^−8^
		13	158,150,159	seq-rs338850979	T/C	0.052	2.56 × 10^−8^
		14	66,511,894	seq-rs80790167	T/G	0.056	9.32 × 10^−8^
		15	57,776,636	seq-rs699198332	A/G	0.063	8.52 × 10^−8^

^1^ Alleles: major/minor allele. ^2^ Minor allele frequency.

## Data Availability

The data presented in this study are available on request from the corresponding author. The data are not publicly available due to commercial interests.

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
