# Peer review of "Genome-Wide Association Analysis and Genetic Parameters for Feed Efficiency and Related Traits in Yorkshire and Duroc Pigs"

_animals, 2022, doi:10.3390/ani12151902_

Round 1

Reviewer 1 Report

The authors have addressed my comments, I just recommend the authors clarify the feed efficiency traits and their component traits in the whole manuscript. 

Line 40: I would recommend the title change “feed efficiency-related traits” to “feed efficiency and related traits”, the authors also might apply it in the whole manuscript (lines 57, 65, 144, 343, 346, 394, 424, 425 659, 673, etc. ) since they performed GWAS for feed efficiency traits (RFI and FCR) and related traits (ADG, Age at 100kg, BF and DFI).

Line 50: the authors might remove “four FE-related traits, including”

Line 108: the sentence “FE was evaluated by four traits… ” is not informative, FE was evaluated based on RFI, FCR, and to some extent residual gain, Residual intake and gain, Kleiber ratio, and partial efficiency of growth (PEG) (https://pubmed.ncbi.nlm.nih.gov/21890504/;  https://www.frontiersin.org/articles/10.3389/fgene.2022.903733/full). Please correct the information.

In references, the journal names should be used in the abbreviation forms.

Line 243: I would recommend the authors cite  Aguilar et al 2010 as well, for their credit for developing the ssGBLUP at the same time. 

Line 412: please check if it is a t-test or chi-squared test

Author Response

The authors have addressed my comments, I just recommend the authors clarify the feed efficiency traits and their component traits in the whole manuscript.

We sincerely thank again your comments and appreciation of our revised manuscript.

Point 1: Line 40: I would recommend the title change “feed efficiency-related traits” to “feed efficiency and related traits”, the authors also might apply it in the whole manuscript (lines 57, 65, 144, 343, 346, 394, 424, 425 659, 673, etc. ) since they performed GWAS for feed efficiency traits (RFI and FCR) and related traits (ADG, Age at 100kg, BF and DFI).

Response 1: Thanks for pointing this out. We have changed “feed efficiency-related traits” or “FE-related traits” to “FE and related traits” in the whole manuscript.

Point 2: Line 50: the authors might remove “four FE-related traits, including”

Response 2: Thanks for the suggestion. We have revised the sentence.

Point 3: Line 108: the sentence “FE was evaluated by four traits… ” is not informative, FE was evaluated based on RFI, FCR, and to some extent residual gain, Residual intake and gain, Kleiber ratio, and partial efficiency of growth (PEG) (https://pubmed.ncbi.nlm.nih.gov/21890504/;  https://www.frontiersin.org/articles/10.3389/fgene.2022.903733/full). Please correct the information.

Response 3: Thank you very much for pointing out, we have modified it.

Point 4: In references, the journal names should be used in the abbreviation forms.

Response 4: It was our mistake. We have changed the Citations Style to MDPI preset in bibliography software package.

Point 5: Line 243: I would recommend the authors cite  Aguilar et al 2010 as well, for their credit for developing the ssGBLUP at the same time.

Response 5: We thank the reviewer for this viewpoint and we agree. We have added it.

Point 6: Line 412: please check if it is a t-test or chi-squared test.

Response 6: We are not sure if you are referring to the test for genetic correlations, as the main text in the revised manuscript submitted is no longer than 400 lines. We performed chi-squared test with one degree of freedom using genetic correlation estimates and its SE to study whether the genetic correlation is different from 0. We have added a description to the Methods section.

Reviewer 2 Report

I checked with careful attention the resubmitted paper by the authors. I see that they put efforts in changing certain aspects, reporting new values in the tables that add additional doubts about the data and how they were previously handled/analysed.

The fundamental questions about this paper remain the same. The experimental design is very poor, the number of samples and animals are low and the types of analyses are not sound.

This paper is for the referee not yet a full story with sound results to be recommended for publication. 

Author Response

I checked with careful attention the resubmitted paper by the authors. I see that they put efforts in changing certain aspects, reporting new values in the tables that add additional doubts about the data and how they were previously handled/analysed.

Response: Yes, we have made extensive revisions to the manuscript based on explicit comments from another reviewer. The update of the values in the tables is mainly due to two reasons. One is to add the hypothesis test of genetic correlation, and the other is to change the genetic parameter estimation model from ssGBLUP to pedigree-based BLUP. These are revisions based on comments from another reviewer and accepted by the reviewer after the manuscript was resubmitted. If you have different suggestions on these contents, please do not hesitate to let us know. We sincerely appreciate your valuable comments and suggestions, which helped us to improve the quality of the article.

Point 1: The fundamental questions about this paper remain the same. The experimental design is very poor.

Response 1: Two pure breeds, Yorkshire and Duroc, which are widely farmed around the world, were selected for this study. All records required for the calculation of feed efficiency traits were collected using automated feeding stations while keeping environmental factors as consistent as possible. Individuals with valid phenotype records are then genotyped, and statistical models are used to identify SNPs associated with the traits. This is a routine experimental design using GWAS to identify feed efficiency-associated SNPs (10.3389/fgene.2020.00692; 10.15389/agrobiology.2019.4.705eng; 10.1371/journal.pone.0183244).

Point 2: the number of samples and animals are low.

Response 2: The number of samples in this study is relatively small, because the current price of performance testing stations is still relatively high, and the test cycle of feed efficiency traits is relatively long, so the cost of obtaining phenotypic records is high. In fact, the sample size of most relevant studies on feed efficiency traits is relatively small. The sample size in the study of Fu et al. (10.3389/fgene.2020.00692) is 296. The sample size in Wang and kadarmindeen's study (10.3390/metabo10050201) was 108. The sample size in the study of ramayo Caldas et al. (10.1186/s12711-019-0490-6) was 350. It can be seen that it is difficult to obtain the information of feed efficiency traits and genotypes. Because of this, we need to study feed efficiency traits to prove that they are heritable and can be improved by breeding methods, so that breeding enterprises are willing to invest money in breeding these traits.

Point 3: the types of analyses are not sound.

Response 3: In the process of analysis, we tried to use different response variables for GWAS, such as the raw phenotypes, phenotypes adjusted for all effects other than genetic, gebv and debv. We also tried to use ssGWAS to identify the causal variants but found that the results of debv were more reliable. If you believe that our analysis methods are still not sound, please don't hesitate to tell us. We would appreciate it if you could give relatively detailed revisions. We will further improve it according to your op suggestions, so that our manuscript can meet your requirements.

Point 4: This paper is for the referee not yet a full story with sound results to be recommended for publication.

Response 4: The content of the analysis in this study already covers the aspects involved in conventional GWAS studies. At the same time, the estimation of genetic parameters can also provide useful information for revealing the genetic structure of FE traits. Considering the possibility of false positive of SNPs associated with traits identified by statistical models, further molecular experiments or verification in a larger population are needed. This is also the content of our next research plan. But these belong to the research content of post-GWAS.

Reviewer 3 Report

Li et al., investigated the heritability of feed efficiency traits using two pig breeds and selected some potential feed efficiency candidate genes. This study provides some useful information for pig breeding. However, there are some concerns before acceptance. Some revisions needed are listed below:

Diets and environmental factors also have effects on gene expression which further affects the phenotype. Therefore, more information related to management and feeding should be provided. Dietary formula should be provided. When did you weigh the pig, for example, in the morning or before feeding? How long is the test period? Did the pigs used in this experiment have some health issues? All relevant information should be provided. 

Which sample (blood or ear tissue) did you use for sequencing? How did you collect the samples. Please clarify.

Please clarify why do you select the body weight between 30-100kg?

Have you further verified the candidate genes in different tissues of the pigs with different feed efficiency? Does sex play a role in regulating feed efficiency. In other words, are the potential candidate genes the same between male and female? 

Meat quality and disease-resistant traits should also be considered during breeding. To your knowledge, do you think high feed efficiency has some negative effects on these two traits mentioned above? 

Author Response

Li et al., investigated the heritability of feed efficiency traits using two pig breeds and selected some potential feed efficiency candidate genes. This study provides some useful information for pig breeding. However, there are some concerns before acceptance. Some revisions needed are listed below.

Reply: We sincerely appreciate your valuable comments and suggestions, which helped us to improve the quality of the article.

Point 1: Diets and environmental factors also have effects on gene expression which further affects the phenotype. Therefore, more information related to management and feeding should be provided. Dietary formula should be provided. When did you weigh the pig, for example, in the morning or before feeding? How long is the test period? Did the pigs used in this experiment have some health issues? All relevant information should be provided.

Response 1: Yes. Environmental factors such as gender, diet, temperature and humidity can all impact which of an animal's genes are expressed, which ultimately affects the animal's phenotype. In order to reduce the impact of environmental factors, phenotypes in our study only recorded from boars and we keep the environmental conditions consistent as far as possible. Because all boars are fed the same feed, we do not provide dietary formula. As we described in Appendix A, feed was given through a single feeder and individuals can feed at any time. All individuals in this study remained in good health during the testing period. Some necessary descriptions have been supplemented in Appendix A.

Point 2: Which sample (blood or ear tissue) did you use for sequencing? How did you collect the samples. Please clarify.

Response 2: Thanks for pointing this out. The relevant description has been supplemented in Appendix B.

Point 3: Please clarify why do you select the body weight between 30-100kg?

Response 3: Data collection in this weight range is mainly to improve the number of effective records and data accuracy. Because individuals were selected at around 30kg rather than at a lower body weight, they were more likely to remain healthy throughout the testing period. However, due to the limitation of the width of the feeder, we cannot select too small piglets, since it is difficult to ensure the accuracy of the weight record.

Point 4: Have you further verified the candidate genes in different tissues of the pigs with different feed efficiency? Does sex play a role in regulating feed efficiency. In other words, are the potential candidate genes the same between male and female?

Response 4: We have tried to search the database (http://iswine.iomics.pro/pig-mrna/mrna/index) for the expression of candidate genes in different tissues of pigs (results not shown). But no relevant molecular verification has been carried out. Further studies are recommended to validate these candidate genes. Phenotypes in our study only recorded from boars, so it was not possible to study whether sex plays an important role in feed efficiency.

Point 5: Meat quality and disease-resistant traits should also be considered during breeding. To your knowledge, do you think high feed efficiency has some negative effects on these two traits mentioned above?

Response 5: Compared with feed efficiency (FE) traits, meat quality traits are more difficult to obtain. Therefore, there are few literatures on the genetic association between FE and meat quality traits. Rauw et al. (DOI: 10.37496/rbz4920200009) found that feed efficiency and body weight gain may be improved in the traditional Iberian production system, with a positive effect on cooking loss but potentially resulting in paler meat. Horodyska et al. (https://doi.org/10.1186/s12864-018-5175-y) found that pigs with high FE had a minor impairment in the quality of meat, in relation to texture and water-holding capacity, and had lower intramuscular fat content. However, higher FE might improve nutritional status in terms of fatty acid composition. For disease-resistance traits, no articles were retrieved to study the relationship between them and FE. However, it can be predicted that individuals with better disease resistance should have higher feed efficiency.

Round 2

Reviewer 2 Report

Previous reports 

This manuscript is a resubmission of an earlier submission. The following is a list of the peer review reports and author responses from that submission.

Round 1

Reviewer 1 Report

Li et al performed GWAS and reported some candidate genes for feed efficiency and component traits in pigs. The sample size is reasonable, and the methods are suitable. I wonder why the authors used the H matrix for estimating genetic parameters. Some other comments below might help.

Line 12-14: DFI and ADG are not really FE traits, I think the better terms for them are FE-related traits or FE component traits.

feed conversion rate might change to feed conversion ratio

Line 15: what are these growth traits?

Not clear “The results revealed the genetic relationship”

Line 19: Genome selection might change to genomic selection or genome-based selection

Line 22: The authors should change unclear to little known or remove unclear. I believe the genetic architecture of FE is quite widely studied.

Line 27: Change Also to Moreover,

Line 28: should check the significant or not significant, the word clearly does not make sense.

Line 26-28: add numbers of SNPs and animals used in GWAS.

Line 32, 36, 37: change feed efficiency to the abbreviation as FE

Gene names should be in Italic

Line 54-55: The authors might provide more comprehensive GWAS results for pigs, not using the information for cattle. At least the authors should summarize how many GWAS have been done for feed efficiency in pigs. At least below GWASs for FE but should be more.

https://www.frontiersin.org/articles/10.3389/fgene.2020.00692/full

https://www.sciencedirect.com/science/article/pii/S1751731119000910

https://www.frontiersin.org/articles/10.3389/fgene.2014.00307/full

https://link.springer.com/article/10.1186/1471-2156-15-27

https://gsejournal.biomedcentral.com/articles/10.1186/s12711-019-0490-6

https://journals.plos.org/plosone/article?id=10.1371/journal.pone.0173482

https://www.frontiersin.org/articles/10.3389/fgene.2018.00220/full

https://link.springer.com/article/10.1007/s00438-017-1325-1

Line 79-81: Why did the authors use the ssGBLUP for the genetic parameters, it should be done using a simple animal model.

Line 97: The authors should change to e ∼ N(0,Iσ2e), for the assumption of error, it should be sigma σ2e, not σ2a.

The authors should add SE for genetic correlations, and might also check if the correlations are significant or not.

Line 118-119: This sentence does not have any information. “We considered four feed efficiency traits and two growth traits, including ADFI, ADG, FCR, RFI, BF, and AGE.”

Line 152: 3.3. genome change to 3.3. Genome

Line 196-197: The authors might change “At an FDR equal 0.05, a total of 100 KEGG pathways or GO terms were significantly associated with FE-related traits in this study.”  To  “A total of 100 KEGG pathways or GO terms were significantly enriched  (FDR < 0.05) for candidate genes of FE-related traits in this study”, and should specify how many KEGGS and how many GO terms.

Line 295: Did the authors use dam in this study, otherwise not necessary to list this information Adam = -9.440

Line 232-234: How did the authors get the H matrix?

Did the authors have information about the pen effects? How many pigs are in a pen?     

 For the whole manuscript, the authors should write feed efficiency or use the abbreviation.

Figures 1 and 2. Why did the authors add a color-coding bar close to the Manhattan plot?

Reviewer 2 Report

This paper treated an interesting research topic. However, there is no novelty in this manuscript. There was no description of the experimental design, and the statistical model as applied is subject to several inaccuracies. The feeding of the animal is not developed

The discussion is very weak, with several generalities.

The targeted breeds were hugely studied using similar methods, and the authors ignored the available literature.

The conclusions of the authors are not supported by the current knowledge in the field.